# In-House Validation of Multiplex PCR for Simultaneous Detection of Shiga Toxin-Producing *Escherichia coli*, *Listeria monocytogenes* and *Salmonella* spp. in Raw Meats

**DOI:** 10.3390/foods11111557

**Published:** 2022-05-25

**Authors:** Chanokchon Jaroenporn, Wannakarn Supawasit, Damkerng Bundidamorn, Pathima Udompijitkul, Anunchai Assawamakin, Sudsai Trevanich

**Affiliations:** 1Department of Food Science and Technology, Faculty of Agro-Industry, Kasetsart University, Bangkok 10900, Thailand; chanokchon.jar@ku.th (C.J.); dif_fai@hotmail.com (W.S.); damkerngb@gmail.com (D.B.); pathima.u@ku.th (P.U.); 2Center for Advanced Studies for Agriculture and Food, Institute for Advanced Studies, Kasetsart University, Bangkok 10900, Thailand; 3Department of Pharmacology, Faculty of Pharmacy, Mahidol University, Bangkok 10400, Thailand; anunchai.asa@mahidol.ac.th

**Keywords:** Multiplex PCR, Shiga toxin-producing *Escherichia coli*, *Listeria monocytogenes*, *Salmonella* spp., validation

## Abstract

The aim of the study was to perform in-house validation of the developed multiplex PCR (mPCR)-based alternative method to detect Shiga toxin-producing *Escherichia coli* (STEC), *Listeria monocytogenes* (*L. monocytogenes*) and *Salmonella* spp. in raw meats following the ISO 16140-2: 2016. A comparative study of the developed mPCR against the Bacteriological Analytical Manual (BAM) method was evaluated for inclusivity and exclusivity, sensitivity and the relative level of detection (RLOD). Inclusivity levels for each target bacterium were all 100%, while exclusivity for non-target bacteria was 100%. The sensitivity of the developed mPCR was calculated based on the analysis of 72 samples of raw meat. The sensitivity of the developed mPCR was 100%. The RLOD values of the developed mPCR for STEC, *L. monocytogenes* and *Salmonella* spp. were 0.756, 1.170 and 1.000, respectively. The developed mPCR showed potential as a tool for the fast, specific and sensitive detection of the three bacteria in the raw meat industry

## 1. Introduction

Shiga toxin-producing *Escherichia coli* (STEC), *Listeria monocytogenes* and *Salmonella* spp. have been the most significant foodborne pathogenic bacteria having a worldwide impact on both consumer health and the economy [1]. A wide variety of poultry and raw meat products have been commonly reported as main sources of these three pathogens [2,3,4]. For reducing the risk of infection for consumers, one microbiological control measure by testing the presence of specific target bacteria in food product is generally done using the standard culture-based method. However, the standard culture method for detection of foodborne pathogenic bacteria is generally expensive, laborious and time-consuming, so many alternative methods including conventional polymerase chain reaction (PCR) have been increasing a great attention [5]. Especially, multiplex PCR assay (mPCR) has many advantages over common PCR as the technique is a cost-effective assay that gives a rapidly simultaneous detection for several microbial targets and improves the accuracy of the identification [6]. However, prior to implementing the developed mPCR assay as an analytical tool, specificity, sensitivity and limit of detection of the developed method must be thoroughly evaluated to show the accuracy of the assay. Some previous mPCR and multiplex RT-PCR assays for simultaneous detection of *E. coli* O157, *L. monocytogenes* and *Salmonella* spp. have been developed [7,8,9,10,11], while only one of the mRT-PCR assays for simultaneous detection of STEC, *L. monocytogenes* and *Salmonella* spp. was recently reported [12]. Nevertheless, extensive validation of those developed mPCR and mRT-PCR assays has not been conducted. There are some differences in the international standard procedures for validation of the developed method, such as International Organization for Standardization 16140-2; 2016-Microbiology of the food chain-Method validation-Part 2; Protocol for the validation of alternative (proprietary) methods against a reference method; [13] and the AOAC Research Institute Performance Tested Method^SM^ program and the AOAC International Methods Committee Guidelines for Validation of Microbiological Methods for Food and Environmental Surfaces [14]. Such international standards are used to prove that the performance characteristics of a newly developed method are at least equivalent or better than those of a standard method. In addition, few publications have reported on ISO 16140-2: 2016-based validations of different alternative methods to detect only single target bacterium in various food samples [15,16,17].

The present study performed an in-house qualitative validation of our developed mPCR to specifically detect STEC, *L. monocytogenes* and *Salmonella* spp. in raw meats. The validation is focused on method comparison study according to ISO 16140-2: 2016. To our knowledge, there are no previous publications in literature investigating an in-house validation study of the developed mPCR assay for simultaneous detection of STEC, *L. monocytogenes* and *Salmonella* spp. as an alternative to the BAM method.

## 2. Materials and Methods

### 2.1. Bacterial Strains and Culture Conditions

In total, 237 bacterial strains were used (Table 1 and Table 2). These strains were purchased from the American Type Culture Collection (ATCC; Manassas, VA, USA); the Centers of Disease Control and Prevention (CDC, Atlanta, GA, USA); the Department of Medical Sciences (DMST; Bangkok, Thailand); and the Department of Microbiology, Faculty of Science, Songkla University (Thailand). All bacterial strains were maintained as frozen stock culture supplemented with 20% glycerol at −20 °C in a freezer. Before use, each stock culture was prepared by activating thrice in tryptic soy broth (TSB) under appropriate incubation conditions, according to [12].

### 2.2. DNA Extraction and the Optimized Multiplex PCR Conditions

DNA extraction and mPCR assay were performed according to [12,18,19,20,21,22,23,24] with a few modifications. Briefly, 1 mL of each pure culture solution or enriched food sample in modified simultaneous enrichment medium (mSEB) containing 1.7% tryptone peptone (Becton Dickinson, Sparks, MD, USA), 0.3% bacto-soytone (Becton Dickinson, Sparks, MD, USA), 0.50% K_2_HPO_4_ (Merck, Darmstadt, Germany), 3% NaCl (Merck, Darmstadt, Germany) and 0.05% yeast extract (Merck, Darmstadt, Germany) after incubation at 37 °C for 24 h was centrifuged at 12,000× *g* for 3 min (Spectrafuge Labnet model M16, Labnet International, Inc., Woodbridge, NJ, USA). After removal of the supernatant, the resulting pellets were washed by resuspension in 1 mL of 0.85% sterile sodium chloride and then centrifuged at 12,000× *g* for 3 min. The obtaining supernatant was discarded. Then, 100 µL of sterile molecular biology-grade water and 0.00333 mg of lysis beads (Bio-Rad Laboratories, Inc., Hercules, CA USA) were added and mixed thoroughly using a cell disruptor (Vortex mixer GENIE 2, Scientific Industries, Inc., Bohemia, NY, USA) for 3 min. The mixture was boiled at 100 °C for 15 min in a heat box (Labnet model D1200, Lanet International, Inc., Woodbridge, NJ, USA) and centrifuged at 12,000× *g* for 3 min. The supernatant was then used as a DNA template for further mPCR assay. The list of primers used, target genes, target strains and the product sizes were shown in Table 3. For preparation of the mPCR reaction, each reaction mixture contained the following components: 1× PCR buffer, 300 µM of dNTP, 2.5 U of *Taq* DNA polymerase, 3.0 mM of magnesium chloride, primers (0.05 μM stx1, 0.05 μM stx2, 0.02 μM eae, 0.02 μM invA, 0.6 μM hlyA and 0.03 μM Internal Amplification Control (IAC), 0.5 ng/µL pUC19 template, 3 µL DNA template and sterile molecular biology-grade water to achieve the total desired volume of 50 µL. The mPCR reaction was performed in a thermo cycle machine (Swift maxi, ESCO Micro Pte Ltd., Singapore) under the following cycling conditions: 1 cycle at 95 °C for 2 min; 35 cycles at 95 °C for 30 s; 57.6 °C for 30 s; 72 °C for 30 s; and a final extension at 72 °C for 7 min. The mPCR products were analyzed on 2.5% agarose gel (Vivantis, Shah Alam, Malaysia) in 1× tris-borate-EDTA buffer. The agarose gel was stained with ethidium bromide and observed using a UV transilluminator (UVP Benchtop, Fisher Scientific, Waltham, MA, USA).

### 2.3. Qualitative Validation Study

The qualitative validation was adapted from ISO 16140-2: 2016 with the focus being on a method comparison that comprised three parts: inclusivity/exclusivity study, sensitivity study and relative level of detection study. The qualitative validation of the developed mPCR assay (alternative method) against the reference method (Bacteriological Analytical Manual) [25,26,27] was conducted at the Microbial Food Safety Laboratory, Department of Food Science and Technology, Faculty of Agro-industry, Kasetsart University, Bangkok, Thailand.

#### 2.3.1. Inclusivity and Exclusivity Studies

Inclusivity and exclusivity testing were performed to evaluate the specificity of the developed mPCR assay for rapid and sensitive detection of three bacterial targets (STEC, *L. monocytogenes* and *Salmonella* spp.). The lists of the target pathogenic bacterial strains and non-target bacterial strains (non-STEC, non-*L. monocytogenes* and non-*Salmonella* spp.) used for inclusivity and exclusivity were shown in Table 1 and Table 2. All pure culture strains tested for the inclusivity study were grown in non-selective TSB under optimal conditions, while each non-target culture strain for the exclusivity study was cultured in an appropriate growth medium under optimal conditions according to the manufacturer’s instructions to achieve high cell populations in the stationary phase [12,28]. The concentrations of each inoculum used for the inclusivity and exclusivity studies was 100 times higher than the minimum detection level of the developed mPCR assay (LOD = 1 CFU/25 mL). The preliminary experiments were performed to investigate the LOD of the developed mPCR assay. When the mixtures of the three representatives of target bacteria (*E. coli* CDC 03-3014: *L. monocytogenes* DMST 17303: *S. Enteritidis* DMST 15676) at ten different combinations and concentrations (1:0:0, 0:1:0, 0:0:1, 1:1:0, 1:0:1, 0:1:1, 1:1:1, 10:10:10, 10^2^:10^2^:10^2^ and 10^3^:10^3^:10^3^ CFU/25 mL, respectively) were initially inoculated into mSEB and incubated at 37 °C for 24 h; the three target bacteria could be detected using the developed mPCR assay. From these preliminary results, the LOD of the developed mPCR assay was 1 CFU/25 mL. Therefore, a tested culture level of approximately 10^2^ CFU/mL was prepared in 25 mL of Butterfield’s phosphate-buffered dilution water and then transferred directly into 225 mL of mSEB. The inoculum suspension was cultured at 37 °C for 24 h. One milliliter of the culture in enriched mSEB was subjected to DNA extraction, as mentioned previously. The DNA obtained was used as a template for the developed mPCR assay as described in 2.2. When doubtful results (false negative or false positive results) from the developed mPCR assay were obtained, the test was repeated using the BAM method.

The results of inclusivity and exclusivity were calculated using the following formulas: % inclusivity = (the number of target bacterial strains testing positive in each group/the number of target bacterial strains tested in each group) × 100; and % exclusivity = (the number of non-target bacterial strains testing negative/the number of non-target bacterial strains tested) × 100.

#### 2.3.2. Sensitivity Study

The sensitivity study was performed to evaluate the developed mPCR assay to detect three target bacteria in three different types of raw meats for comparison with the BAM method as a reference. In total, 72 unpaired raw meat samples consisting of raw pork (*n* = 24), raw chicken (*n* = 24) and raw beef (*n* = 24) were purchased from different retail markets in Bangkok, Thailand. All 72 unpaired raw meats were separated into a non-contaminated group and an artificially contaminated (spiking) group with unstressed or chill-stressed cells of each representative target bacterium (*E. coli* O26:H11 CDC 03-3014, *L. monocytogenes* DMST 23,146 and *Salmonella* Enteritidis DMST 16813) in the form of triple inoculation to give a final concentration of each target bacterium at 1 cell/25 g of unpaired raw meat. Chill-stressed cells were prepared from unstressed cells grown thrice in appropriate non-selective broth with three different continuously incubation conditions (37 °C for 24 h, 37 °C for 24 h or 37 °C for 18 h). In order to prepare chill-stressed cells with sufficient stress efficiency at 0.5 log CFU/mL difference evaluated by counting the cell number on both of selective and non-selective media, the unstressed cells at the level of 10^4^ CFU/mL were kept at 4 ± 1 °C for 14 days. Furthermore, low and high background microbiota in unpaired raw meats were included in both the non-contaminated and artificially contaminated groups. The average number of aerobic plate counts (10^2^–10^3^ CFU/g) found in the initial raw meat samples was used as the condition of low background microbiota. For the preparation of unpaired raw meat samples with high background normal biota, *Pseudomonas aeruginosa* DMST 4739 was used as representative of normal biota in raw meats [29,30,31,32] and the appropriate concentration of the culture was directly inoculated into the food sample to reach a final population of high background microbiota at approximately 4.0 × 10^5^ CFU/g.

For preparation of unpaired raw meat samples, 1.5 kg of each unpaired raw meat sample purchased from approved manufacturers was cut and divided into two sets of 50 g test portions and eight sets of 25 g test portions. Two of the 50 g test portions were analyzed for aerobic plate count [33]. Three of the 25 g test portions were checked using the BAM method [25,26,27] to ensure none of STEC, *L. monocytogenes* and *Salmonella* spp. were present before use, while one of the 25 g test portions was confirmed for the presence/absence of the three target bacteria by the developed mPCR assay. The remining four 25 g test portions were used for the sensitivity test. Three out of four samples of 25 g test portions were individually examined for STEC, *L. monocytogenes* and *Salmonella* spp. while the remaining sample of 25 g test portion was analyzed for the presence/absence of the three target bacteria using the developed mPCR assay. According to ISO 16140-2: 2016 for interpretation of the sample results between the reference method (the BAM method) and alternative method (the developed mPCR method) for an unpaired study, additional confirmation of the developed mPCR assay result shall be done by using the BAM method for each target bacterium (namely, the confirmed mPCR result using the BAM method).

The sensitivity of the developed mPCR assay (SE_alt_), sensitivity of the BAM method (SE_ref_), relative trueness (RT) and the false positive ratio for the developed mPCR assay (FPR) were calculated according to ISO 16140-2: 2016 as follows:
(1)$${\mathrm{SE}}_{\mathrm{alt}}=\frac{\left(PA+ND\right)}{\left(PA+ND+PD\right)}\text{×}100,\;{\mathrm{SE}}_{\mathrm{ref}}=\frac{\left(PA+ND\right)}{\left(PA+ND+PD\right)}\text{×}100,\;\mathrm{RT}=\frac{\left(PA+ND\right)}{N}\text{×}100\;\mathrm{and}\;\mathrm{FPR}=\frac{FP}{NA}\text{×}100$$where PA is positive agreement, PD is positive deviation, NA is negative agreement, ND is negative deviation and N is the total number of samples.

#### 2.3.3. Relative Level of Detection Study

The relative level of detection (RLOD) value of the developed mPCR assay against BAM was determined for detection of each target bacterium in raw duck meat samples as ISO 16140-2: 2016 recommended that the type of food sample used for RLOD study should be different from those used in the sensitivity study in order to have a better representation of the evaluated category. In total, 30 unpaired samples of raw duck meat were prepared with three different levels of contamination of the three representative target bacteria. Five samples were uninoculated (0 CFU/25 g). Twenty samples were artificially contaminated with a low level of each target bacteria at 0.5 CFU/25 g and five samples were contaminated with a high level of each target bacteria at 1.5 CFU/25 g. Each 1.5 kg sample of unpaired raw duck meat purchased from approved manufacturers was cut and divided into eight sets of 25 g test portions. Four of 25 g test portions were checked using the BAM method [25,26,27] and the developed mPCR assay to ensure none of STEC, *L. monocytogenes* and *Salmonella* spp. were present before use. The other four of the 25 g test portions were used for RLOD test. Three out of four samples of 25 g test portions were individually examined for STEC, *L. monocytogenes* and *Salmonella* spp. using the BAM method while the remaining sample of the 25 g test portions was analyzed for the presence/absence of the three bacterial targets using the developed mPCR assay and confirmed mPCR assay with the BAM method. For preparation of the culture, each representative of the target bacteria used in the sensitivity study was prepared in specific enrichment media and incubated under specific conditions as described previously. In addition, RLOD values were calculated using the Excel^®^ spreadsheet which was freely accessed at http://standards.iso.org/iso/16140 (accessed on 4 January 2022).

## 3. Results

### 3.1. Inclusivity and Exclusivity

The inclusivity of mPCR assay was performed with 207 different isolates of the three target bacterial strains, covering 55 STEC strains, 102 *Salmonella* serovars and 50 *L. monocytogenes* strains. The inclusivity results are shown in Table 4 and Appendix A Appendix A. All STEC and *L. monocytogenes* strains as well as *Salmonella* serovars that were tested produced positive results with 100% specificity.

All 30 strains of non-target bacteria were tested in the exclusivity study. None of the non-target bacterial strains were positive with the developed mPCR assay (Table 4 and Appendix A Appendix A;). The inclusivity and exclusivity results are summarized in Table 5. The internal control amplification (IAC) was amplified in every run of mPCR reactions.

### 3.2. Sensitivity Study

The raw meat chicken, pork and beef samples were analyzed with the developed mPCR assay and the BAM method for each target bacteria and then verified by the confirmed mPCR result using the BAM method. All 36 artificially contaminated unpaired sample of raw meat and all 36 non-contaminated unpaired raw meat samples were correctly analyzed by both the developed mPCR assay and the BAM method. Table 6 presents the results of the unpaired sensitivity study of the developed mPCR assay and BAM method to detect the three target bacteria in the three different types of raw meat sample.

### 3.3. Relative Level of Detection Study

All five non-contaminated raw duck meat samples and the five high level contaminated raw duck meat samples were correctly detected by the developed mPCR assay and BAM method while some low-level contaminated raw duck meat samples produced identical positive results from both methods (Table 7). For the low-level contaminated samples, the developed mPCR assay showed positive results for 12, 8 and 11 out of 20 samples for the presence of representative STEC, *L. monocytogenes* and *Salmonella* spp., respectively. Furthermore, 10, 9 and 11 out of 20 samples were confirmed positive for representative STEC, *L. monocytogenes* and *Salmonella* spp., respectively, by the BAM method. The RLODs of the developed mPCR assay to detect each representative STEC, *L. monocytogenes* and *Salmonella* spp. were 0.756, 1.170 and 1.000, respectively, with a lower limit of the 95% confidence interval (CI) of 0.314 and an upper limit of the 95% CI of 1.823, a lower limit of the 95% CI of 0.437 and an upper limit of the 95% CI of 3.132 and a lower limit of the 95% CI of 0.447 and an upper limit of the 95% CI of 2.240, respectively.

## 4. Discussion

Alternative methods based on the mPCR assay which used primer sets of stx1, stx2, eae, invA and hlyA, have been developed for rapid and specific detection of STEC, *L. monocytogenes* and *Salmonella* spp. in raw meat samples. However, the developed mPCR assay needed to be validated before implementation. Based on our knowledge, there has been no mPCR assay for simultaneous detection of STEC, *L. monocytogenes* and *Salmonella* spp. which are validated according to ISO 16140-2: 2016 and compared to the BAM method. Therefore, this is the first report of validation of the developed mPCR assay for detection of those three target bacteria in raw meat samples. This qualitative validation was partially conducted following ISO 16140-2: 2016 as an unpaired trial study. This means that the test portion of food sample and enrichment broth used were different between the alternative method and reference method. Similar to other works conducted by some researchers who had resource limitations in the organizing laboratory [16,34,35], it was only possible within this present study to use one food category (raw meat) with three different types of raw meat (raw pork, raw chicken and raw beef) for the qualitative validation. The validation of the developed mPCR assay compared to the BAM method mainly focused on using a comparative study which included inclusivity, exclusivity, sensitivity and RLOD studies of the raw meat samples.

The results of inclusivity testing showed the developed mPCR assay had 100% specific detection for the 55 different strains of STEC, 50 strains of *L. monocytogenes* and 102 different serovars of *Salmonella* (Table 3). The STEC strains used in this study were displayed different combined presence or absence of the *stx1*, *stx2* and/or *eae* genes, which are hallmarks of STEC. The *stx1*, *stx2* and *eae* genes were each found in 28.6%, 83.9% and 82.1%, respectively, of the tested STEC. However, the authors of [36] reported that the development of a detection method for pathogenic STEC strains was challenging and needed to include selection of genetic markers other than only the *stx1*, *stx2* and *eae* genes. For exclusivity testing, none of the non-target bacteria strains tested gave no positive product of each target gene. This indicated an exclusivity value equivalent to 100%, indicating a high specificity of the developed mPCR method for the fast and specific detection of the three target bacteria, with no cross-reactions in non-target bacteria.

In the sensitivity study, the SE_alt_ and RT values of the developed mPCR assay for each target bacterium were all 100%. In addition, the FPR values were 0% due to the absence of false-positive or false-negative detection. ISO 16140-2: 2016 sets the acceptability limit value of ND-PD for an unpaired sensitivity study of one category of food sample at 3. The ND-PD values for each target bacterium were all zero (below the parameter value of 3). Therefore, the developed mPCR assay was within the acceptability limit for sensitivity as an alternative to the BAM method alone for specific detection of STEC, *L. monocytogenes* and *Salmonella* spp. in raw meat. In fact, to closely follow the guidelines for the sensitivity study referred to in ISO 16140-2: 2016, naturally contaminated raw meat samples should be used. However, naturally target bacteria-contaminated raw meat was not available and so artificially contaminated raw meat samples were investigated instead. Studies [34,35] also performed the sensitivity study with artificially contaminated pork meat and chicken neck skin. The high sensitivity (SE_alt_, 100%) and relative trueness (RT, 100%) obtained from the developed mPCR assay suggested that this method has the potential to complement the BAM method for the true detection of each target bacterium in raw meat samples. Obtaining such high values for SE_alt_ and RT, as well as the ND-PD value being within the acceptability limit, are important indicators because these values demonstrate whether the developed method is acceptable for analyzing each target bacterium contaminated in the samples of raw meat matrices. Data obtained from the sensitivity study demonstrated that artificial contamination with non-stressed or chill-stressed cells of representative target bacteria and different levels of background microbiota (high and low background) in raw meat samples did not affect the performance of the developed mPCR assay. Information obtained from various databases indicated that few reports on sensitivity studies following ISO 16140-2: 2016 have focused on *L. monocytogenes* or *Salmonella* spp. The authors of [15] evaluated the effectiveness of ISO 6579 and VIDAS UP *Salmonella* SPT in slaughtered sheep, reporting values of 94.44% for SE_alt_ and 99.79% for RT. Relatively high values of SE_alt_ and RT were reported by [17] who evaluated RT-PCR to detect *L. monocytogenes* contaminated in dry-cured ham. [16] investigated a loop-mediated amplification combined with a standard culturing procedure to detect *Salmonella enterica* in soya meal recording results of 100% for both SE_alt_ and RT.

Since the LOD of the developed mPCR assay was 1 CFU/25 g, the level of detection at 50% values evaluated according to the guidelines prescribed in ISO 16140-2: 2016 were all similar at 0.5 CFU/25 g for each target bacterium. For the RLOD study, ISO 16140-2: 2016 recommends using a minimum of three levels of contamination per type of food sample. In the present study, three levels of contamination of each target bacterium were tested, consisting of 0, 0.5 and 1.5 CFU/25 g of raw duck meat samples. The RLOD results are shown in Table 5. The RL values for the developed mPCR assay for detection of STEC, *L. monocytogenes* and *Salmonella* spp. were 0.76, 1.2 and 1.0, respectively. ISO 16140-2: 2016 sets the acceptability limit of RLOD for unpaired studies at 2.5. Therefore, the RLODs of the developed mPCR assay for each target bacterium complied with the established limits, indicating that the developed mPCR assay was acceptable and can reasonably be considered as good as the BAM method with regard to their respective contamination levels for the detection of the three bacterial targets. Interestingly, the LOD value of the developed mPCR assay for STEC was smaller than the LOD for the BAM method. The results indicated that the developed mPCR assay was likely to detect a lower contamination level of STEC than the BAM method. The numbers of STEC, *L. monocytogenes* and *Salmonella* spp. in raw meats are often low (less than 10-100 CFU/g) and those bacteria may be present in a state of sub-lethal injury because of low temperature storage or the presence of natural antimicrobial agents [37,38,39]. Several publications have recently reported the evaluation of RLOD values for alternative methods according to ISO 16140-2: 2016. For example, [17] evaluated RT-PCR to specifically detect *L. monocytogenes* present in sliced dry-cured ham at three levels of contamination (0, 0.3 and 0.9 CFU/25 g). They obtained an RLOD value of 1.00 which was below the acceptability limit. In addition, an RLOD value of 1.00 was observed during the study by [16] who developed LAMP combined with a pre-enrichment step to detect *Salmonella enterica* in soya meal samples.

## 5. Conclusions

The developed mPCR assay combined with enrichment in mSEB to detect STEC, *L. monocytogenes* and *Salmonella* spp. in raw meats has the potential to be an accurate alternative to the standard BAM method with comparable or better sensitivity and specificity but with the potential advantages of savings in cost, time and labor. However, other food categories and types should be tested to strictly validate conformation to the ISO 16140-2: 2016 guidelines. The next step in validation based on an interlaboratory study for the developed mPCR assay is under investigation to conform to ISO 16140-2: 2016.

## Figures and Tables

**Table 1 foods-11-01557-t001:** Target bacterial strains used for inclusivity of the developed mPCR assay.

Number of Strains	Target Bacterial Strains
55	Shiga toxin-producing *Escherichia coli* (STEC ^a^) DMST ^b^ 48719; STEC DMST 50662; STEC PSU ^c^ 5023; STEC DMST 19341; STEC DMST 50661; STEC PSU 38; STEC PSU 5030; STEC DMST 50660; STEC PSU 4189; STEC PSU 4159; STEC DMST 30538; STEC DMST 30536; STEC DMST 30537; STEC PSU 149; STEC PSU 150; STEC DMST 19340; STEC DMST 19342; STEC DMST 30539; STEC DMST 50659; STEC PSU 133; STEC PSU 135; STEC PSU 148; STEC PSU 4153; STEC PSU 4154; STEC PSU 4155; STEC PSU 4156; STEC PSU 4157; STEC PSU 4158; STEC PSU 4160; STEC PSU 4161; STEC PSU 4162; STEC PSU 4163; STEC PSU 4164; STEC PSU 4165; STEC PSU 4166; STEC PSU 4167; STEC PSU 4169; STEC PSU 4170; STEC PSU 4171; STEC PSU 4172; STEC PSU 4173; STEC PSU 4191; STEC PSU 4192; STEC PSU 4193; STEC PSU 4196; STEC PSU 4197; STEC PSU 5026; STEC PSU 5027; STEC PSU 5028; STEC PSU 5029; STEC PSU 3802; STEC PSU 4190; STEC PSU 4195; STEC PSU 4198 and STEC CDC ^d^ 03-3014
50	*L. monocytogenes* ATCC ^e^ 7644; *L. monocytogenes* Li 23 ATCC 19114 *L. monocytogenes* Li 2107 ATCC 19116; *L. monocytogenes* DMST 41455; *L. monocytogenes* DMST 17303; *L. monocytogenes* DMST 20093; *L. monocytogenes* DMST 20422; *L. monocytogenes* DMST 20423; *L. monocytogenes* DMST 20425; *L. monocytogenes* DMST 21164; *L. monocytogenes* DMST 21165; *L. monocytogenes* DMST 23136; *L. monocytogenes* DMST 23146; *L. monocytogenes* DMST 23150; *L. monocytogenes* DMST 23151; *L. monocytogenes* DMST 23710; *L. monocytogenes* DMST 27738; *L. monocytogenes* DMST 31799; *L. monocytogenes* DMST 31800; *L. monocytogenes* DMST 31801; *L. monocytogenes* DMST 31802; *L. monocytogenes* DMST 31804; *L. monocytogenes* DMST 32862; *L. monocytogenes* DMST 33253; *L. monocytogenes* DMST 36156; *L. monocytogenes* DMST 37884; *L. monocytogenes* DMST 37885; *L. monocytogenes* DMST 41456; *L. monocytogenes* DMST 41457; *L. monocytogenes* DMST 41458; *L. monocytogenes* DMST 41642; *L. monocytogenes* DMST 41647; *L. monocytogenes* DMST 44932; *L. monocytogenes* DMST 44933; *L. monocytogenes* DMST 44934; *L. monocytogenes* DMST 44935; *L. monocytogenes* DMST 44936; *L. monocytogenes* DMST 45433; *L. monocytogenes* DMST 45683; *L. monocytogenes* DMST 45984; *L. monocytogenes* DMST 46456; *L. monocytogenes* DMST 47501; *L. monocytogenes* DMST 47502; *L. monocytogenes* DMST 47503; *L. monocytogenes* DMST 50339; *L. monocytogenes* 100; *L. monocytogenes* 101; *L. monocytogenes* 108; *L. monocytogenes* 310 and *L. monocytogenes* Scott A
102	*S.* Aberdeen DMST 19198; *S.* Abony PVKU 1 ^f^; *S.* Agona DMST 23970; *S.* Alachua DMST 19203; *S.* Albany DMST 50696; *S.* Altona DMST 62226; *S.* Amsterdam WPKU 1 ^g^; *S.* Anatum DMST 50705; *S.* Apeyeme PVKU 2; *S.* Augustenborg DMST 50631; *S.* Bangkok DMST 50834; *S.* Bareilly DMST 62231; *S.* Bergen DMST 19206; *S.* Blockley DMST 16821; *S.* Bongori ATCC 43975; *S.* Bovismorbificans DMST 17379; *S.* Braenderup DMST 62234; *S.* Bredeney PVKU 3; *S.* Canstatt PVKU 4; *S.* Cerro DMST 19200; *S.* Chester WPKU 2; *S.* Chicago KU 1 ^h^; *S.* Corvalis KU 2; *S.* Derby DMST 16880; *S.* Dublin WPKU 3; *S.* Eastbourne WPKU 4; *S.* Emek WPKU 5; *S. enterica* subsp. *Salamae ser*, *17:gt:* DMST 19207; *S.* Adelaide ATCC 10718; *S.* Bispebjerg ATCC 9842; *S.* Choleraesuis ATCC 6958; *S.* Gaminara ATCC 8324; *S.* Heerlen ATCC 15792; *S.* Hillingdon ATCC 9184; *S.* Illinois ATCC 11646; *S.* Inverness ATCC 10720; *S.* Kirkee ATCC 8322; *S.* Oranienburg ATCC 9239; *S.* Pullorum ATCC 9120; *S.* Simsbury ATCC 12004; *S.* Vellore ATCC 15611; *S.* Zwickau ATCC 15805; *S.* Dar-es-salaam ATCC 6959; *S.* Hooggraven ATCC 15786; *S.* Enteritidis DMST 15676; *S.* Falkensee DMST 50716; *S.* Fresno DMST 19197; *S.* Give DMST 50827; *S.* Hadar DMST 32769; *S.* Havana DMST 50710; *S.* Heidelberg PVKU 5; *S.* Hvittingfoss DMST 62220; *S.* Indiana PVKU 6; *S.* Infatis PVKU 7; *S.* I4,12:i:- PVKU 8; *S.* Johnnesburg DMST 50835; *S.* Kedougou DMST 33890; *S.* Kentucky DMST 50701; *S.* KiamBu PVKU 9; *S.* Krefeld DMST 62227; *S.* Lexington DMST 50707; *S.* Liverpool PVKU 10; *S.* Livingstone DMST 50633; *S.* London DMST 62232; *S.* Manhattan PVKU 11; *S.* Matopeni DMST 62218; *S.* Mbandaka DMST 62238; *S.* Minnesota KU 3; *S.* Molade PVKU 12; *S.* Montevideo PVKU 13; *S.* Moscow PVKU 14; *S.* Muenchen PVKU 15; *S.* Muenster DMST 62235; *S.* Newport DMST 15675; *S.* Orion WPKU 6; *S.* Oslo WPKU 7; *S.* Ouakam DMST 50824; *S.* Panama DMST 50703; *S.* Paratyphi A DMST 15673; *S.* Paratyphi B WPKU 8; *S.* Paratyphi C WPKU 9; *S.* Poona KU 4; *S.* Ramat-gan WPKU 10; *S.* Rissen DMST 16876; *S.* Rubislaw DMST 62223; *S.* Saintpaul DMST 62225; *S.* Schwarzengrund WPKU 11; *S.* Typhi WPKU 12; *S.* Singapore DMST 50636; *S.* Stanley DMST 33894; *S.* Tennessee WPKU 13; *S.* Thempson PVKU 16; *S.* Typhimurium DMST 562; *S.* Brunei KU 5; *S.* Virchow DMST 16857; *S.* Wandsworth DMST 19204; *S.* Warthington DMST 33889; *S.* Waycross DMST 19205; *S.* Weltevreden DMST 16820; *S.* Urbana PVKU 17; *S.* Soerenga PVKU 18 and *S.* Weston PVKU 19

^a^ STEC = Shiga toxin-producing *Escherichia coli;*
^b^ DMST = Department of Medical Science, Ministry of Public Health, Bangkok, Thailand; ^c^ PSU = Department of Microbiology, Faculty of Science, Prince of Songkla University, Songkla, Thailand; ^d^ CDC = Centers for Disease Control and Prevention, Atlanta, GA, USA; ^e^ ATCC = The American Type Culture Collection, Manassas, VA, USA; ^f^ PVKU = Collection cultures in our laboratory, Kasetsart University, Bangkok, Thailand; ^g^ WPKU = Collection cultures in our laboratory, Kasetsart University, Bangkok, Thailand; ^h^ KU = Collection cultures in our laboratory, Kasetsart University, Bangkok, Thailand.

**Table 2 foods-11-01557-t002:** Non-target bacterial strains used for exclusivity of the developed mPCR assay.

Number of Strains	Non-Target Bacterial Strains
30	*Aeromonas hydrophila* DMST ^a^ 21250; *Bacillus cereus* DMST 5040; *Campylobacter coli* DMST 18034; *Campylobacter jejuni* DMST 15190; *Citrobacter freundii* DMST 16368; *Cronobacter sakazakii* DMST 17894; *Enterobacter cloacae* DMST 434; *Enterococcus faecalis* DMST 4736; *Klebsiella pneumonia* DMST 8216; *Lactococcus lactis* KU 6 ^b^; *Lactobacillus brevis* KU 7; *Listeria innocua* DMST 9011; *Listeria ivanovii* DMST 9012; *Pediococcus pentosaceus* DMST 18752; *Proteus vulgaris* DMST 557; *Pseudomonas aeruginosa* DMST 4739; *Pseudomonas fluorescens* KU 8; *Shigella dysenteriae* DMST 15111; *Shigella flexneri* DMST 4423; *Shigella sonnei* DMST 561; *Staphylococcus aureus* DMST 8840; *Streptococcus pyogenes* DMST 30653; *Streptococcus suis* serotype II DMST 18783; *Vibrio cholerae* nonO1/nonO139 DMST 2873; *Vibrio parahaemolyticus* DMST 21243; *Vibrio vulnificus* DMST 21245; *Yersinia enterocolitica* DMST 8012; *Yersinia pseudotubercolosis* DMST 16385; *Bacillus subtilis* (Ehrenberg) Cohn BCC ^c^ 6327 and *Staphylococcus epidermidis* KU 9

^a^ DMST = Department of Medical Science, Ministry of Public Health, Bangkok, Thailand; ^b^ KU = Collection cultures in our laboratory, Kasetsart University, Bangkok, Thailand; ^c^ BCC = BIOTECH Culture Collection, Bangkok, Thailand.

**Table 3 foods-11-01557-t003:** Primer pairs and target genes used for the developed mPCR assay.

Primer	Primer Sequence (5′–3′)	Target Gene	Target	Product Size (bp)	Reference
stx1-F stx1-R	TGTAACTGGAAAGGTGGAGTATACA GCTATTCTGAGTCAACGAAAAATAAC	*stx1*	STEC	210	[19]
stx2-F stx2-R	ATCAGTCGTCACTCACTGGT CTGCTGTCACAGTGACAAA	*stx2*	STEC	110	[20]
eae-F eae-R	TCAATGCAGTTCCGTTATCAGTT GTAAAGTCCGTTACCCCAACCTG	*eae*	STEC	482	[21]
hlyA-F hlyA-R	AAATCATCGACGGCAACCT GGACGATGTGAAATGAGC	*hlyA*	*L. monocytogenes*	348	[22]
invA-F invA-R	GTGAAATTATCGCCACGTTCGGGCAA TCATCGCACCGTCAAAGGAACC	*invA*	*Salmonella* spp.	284	[23]
IAC-F IAC-R	CAGGATTGACAGAGCGAGGTATG CGTAGTTAGGCCACCACTTCAAG	*ori*	pUC19	65	[24]

*stx1* and *stx2* gene: Shiga toxin gene; *eae* gene: Intimin gene; *invA* gene: Invasion gene; *hlyA* gene: Hemolysin A gene.

**Table 4 foods-11-01557-t004:** Inclusivity and exclusivity of the developed mPCR assay for specific detection of three target bacteria.

Number of Strains		Bacterial Strains	Test Result
Target bacterial strains	3	Shiga toxin-producing *Escherichia coli* (STEC ^a^) DMST ^b^ 48719; STEC DMST 50662 and STEC PSU ^c^ 5023	Positive for *stx1* gene ^e^ (product size 210 bp)
	4	STEC DMST 19341; STEC DMST 50661; STEC PSU 38 and STEC PSU 5030	Positive for *stx2* gene ^f^ (product size 110 bp)
	3	STEC DMST 50660; STEC PSU 4189 and STEC PSU 4159	Positive for *stx1* and *stx2* genes
	5	STEC DMST 30538; STEC DMST 30536; STEC DMST 30537; STEC PSU 149 and STEC PSU 150	Positive for *stx1* and *eae* ^g^ genes (product size 482 bp)
	35	STEC DMST 19340; STEC DMST 19342; STEC DMST 30539; STEC DMST 50659; STEC PSU 133; STEC PSU 135; STEC PSU 148; STEC PSU 4153; STEC PSU 4154; STEC PSU 4155; STEC PSU 4156; STEC PSU 4157; STEC PSU 4158; STEC PSU 4160; STEC PSU 4161; STEC PSU 4162; STEC PSU 4163; STEC PSU 4164; STEC PSU 4165; STEC PSU 4166; STEC PSU 4167; STEC PSU 4169; STEC PSU 4170; STEC PSU 4171; STEC PSU 4172; STEC PSU 4173; STEC PSU 4191; STEC PSU 4192; STEC PSU 4193; STEC PSU 4196; STEC PSU 4197; STEC PSU 5026; STEC PSU 5027; STEC PSU 5028 and STEC PSU 5029	Positive for *stx2* and *eae* genes
	5	STEC PSU 3802; STEC PSU 4190; STEC PSU 4195; STEC PSU 4198 and STEC CDC ^d^ 03-3014	Positive for *stx1*, *stx2* and *eae* genes
	50	*L. monocytogenes* strains in Table 1	Positive for *hlyA* gene ^h^ (product size 348 bp)
	102	*Salmonella* spp. serotypes in Table 1	Positive for *invA* gene ^i^ (product size 284 bp)
Non-target bacterial strains	30	Non-target bacterial strains in Table 2	Negative

^a^ STEC = Shiga toxin-producing *Escherichia coli;*
^b^ DMST = Department of Medical Science, Ministry of Public Health, Bangkok, Thailand; ^c^ PSU = Department of Microbiology, Faculty of Science, Prince of Songkla University, Songkla, Thailand; ^d^ CDC = Centers for Disease Control and Prevention, Atlanta, GA, USA; ^e^ *stx1* gene = Shiga toxin 1 gene; ^f^
*stx2* gene = Shiga toxin 2 gene; ^g^ *eae* gene = Intimin gene; ^h^ *hlyA* gene = Hemolysis A gene; ^i^
*invA* gene = Invasion protein A gene.

**Table 5 foods-11-01557-t005:** Results of inclusivity and exclusivity study of developed mPCR assay for detection of three target bacteria and non-target bacteria.

Bacteria	Total Number of Isolates Tested	Total Number of Positive Results by the Developed mPCR Assay	Inclusivity/Exclusivity (%)
Shiga toxin-producing *E. coli*	55	55	100
*L. monocytogenes*	50	50	100
*Salmonella* spp.	102	102	100
non-target bacteria ^a^	30	30	100

^a^ Non-target bacteria = non- Shiga toxin-producing *E. coli*, non- *L. monocytogenes* and non- *Salmonella* spp.

**Table 6 foods-11-01557-t006:** Results of each parameter in unpaired sensitivity study of the developed mPCR assay and BAM method for detection of each target bacterium.

Parameter	Value (%)
Sensitivity for developed mPCR assay (SE_alt_)	100
Sensitivity for BAM method (SE_ref_)	100
Relative trueness (RT)	100
False positive ratio (FPR)	0

**Table 7 foods-11-01557-t007:** Relative limit of detection of the developed mPCR assay evaluated against BAM method for detection of each target bacterium in raw duck meats.

Method	Positive Results of Sample Contaminated with Different Levels of Contamination			RLOD ^g^
	0 ^a^	0.5 ^b^	1.5 ^c^	
mPCR for STEC	0/5 ^d^	12/20 ^e^	5/5 ^f^	0.756
BAM ^h^ for STEC	0/5	10/20	5/5	
mPCR for *L. monocytogenes*	0/5	8/20	5/5	1.170
BAM for *L. monocytogenes*	0/5	9/20	5/5	
mPCR for *Salmonella* spp.	0/5	11/20	5/5	1.000
BAM for *Salmonella* spp.	0/5	11/20	5/5	

^a^ no contamination (0 CFU/25 g); ^b^ Low level contamination (0.5 CFU/25 g); ^c^ High level contamination (1.5 CFU/25 g); ^d, e, f^ Total of replicates tested; ^g^ RLOD = Relative limit of detection; ^h^ BAM = Bacteriological Analytical Manual.

## Data Availability

The data presented in this work are available in this article and Appendix A.

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
