# Peer review of "In-House Validation of Multiplex PCR for Simultaneous Detection of Shiga Toxin-Producing Escherichia coli, Listeria monocytogenes and Salmonella spp. in Raw Meats"

_foods, 2022, doi:10.3390/foods11111557_

Round 1
Reviewer 1 Report
This manuscript describes the validation process of a mPCR assay developed by authors for simultaneous detection of Shiga Toxin-producing Escherichia coli, Listeria monocytogenes and Salmonella spp. in raw meats. The aim of the work is interesting as these three pathogens are of great Public Health concern. Moreover, at the moment, there are some mPCR protocols described in bibliography, but most of them have not been validated to demonstrate their usefulness in comparison to standard reference methods. Methodology is adequate and well described. Results are good and conclusions are according to them.
My main concern is that authors do not describe the mPCR protocol that they have developed and are now validating and they do not provide any reference to it either, thus making impossible to reproduce the assays: amount of sample submitted to enrichment, mSEB composition or incubation time for enrichment are not detailed. This is a crucial point, and authors should include it in a new version of their paper.
In this same sense, authors point out that the LOD of the developed mPCR assay was 1 CFU/25 mL (Lines 126 and 319), but they do not describe how they have reached this conclusion.
Some other points:
Abstract (Line 17): As it is the first time that it is mentioned, please provide the complete name of BAM
Material and Methods:
The list of primers, size of amplicons and references (Lines 87 to 98) should better be included in a table rather than in the text
“in an appropriate growth medium” (Lines 122-123): Please, provide the media used or add a reference for them
Please, change “microflora” and “flora” to “microbiota” and “biota” all along the text
“Pseudomonas aeruginosa DMST 4739 was used as representative of normal flora in raw meats” (Line 171): Please, provide a reference or explain why
“none of STEC, L. monocytogenes and Salmonella spp. were present and the developed mPCR assay were present before use.” (Lines 204-205): Please, revise redaction
Results: “The raw meat chicken, pork and beef samples were analyzed with the developed mPCR assay and the BAM method for each target bacteria and then confirmed by mPCR using the BAM method” (Lines 232 to 233): This sentence is quite confusing. Please, explain what do you mean when saying that a result is confirmed by PCR using the BAM method
Author Response
Dear Reviewer 1,
Thank you for your valuable suggestions and questions. Our responses are in the file attached. Please see the attachment.
Best regards,
Sudsai Trevanich

Reviewer 2 Report
Line 15 The abbreviation of Listeria monocytogenes should be indicated in parentheses.
Line 24 Key words should write the full name of “STEC”, “L. monocytogenes”.
Line 35 PCR detection methods (including mPCR) are common. What else is new about the method?
Line 36 Only advantaged are given for mPCR tests, are there no limits? Can the method tell the three bacteria apart?
Line 65 Please specify the species of these bacteria.
Line 140, Line 147 Table 1 and Table 2 are descriptions of results, why put them in the methods part?
Line 222, Please provide part of PCR gel electrophoresis pictures to support test results.
Author Response
Dear Reviewer 2,
Thank you for your valuable suggestions and questions. Our responses are in the file attached. Please see the attachment.
Best regards,
Sudsai Trevanich

Round 2
Reviewer 1 Report
Manuscript has been revised according to previous comments and suggestions. I have no further comment.